# Inhibition of BRD4 Promotes Pexophagy by Increasing ROS and ATM Activation

**DOI:** 10.3390/cells11182839

**Published:** 2022-09-12

**Authors:** Yong Hwan Kim, Doo Sin Jo, Na Yeon Park, Ji-Eun Bae, Joon Bum Kim, Ha Jung Lee, So Hyun Kim, Seong Hyun Kim, Sunwoo Lee, Mikyung Son, Kyuhee Park, Kwiwan Jeong, Eunbyul Yeom, Dong-Hyung Cho

**Affiliations:** 1BK21 FOUR KNU Creative BioResearch Group, School of Life Sciences, Kyungpook National University, Daegu 41566, Korea; 2Brain Science and Engineering Institute, Kyungpook National University, Daegu 41566, Korea; 3Orgasis Corp., Suwon 16229, Gyeonggi-do, Korea; 4Bio-Center, Gyeonggido Business & Science Accelerator, Suwon 16229, Gyeonggi-do, Korea

**Keywords:** pexophagy, peroxisome, molibresib, BRD4, ROS

## Abstract

Although autophagy regulates the quality and quantity of cellular compartments, the regulatory mechanisms underlying peroxisomal autophagy (pexophagy) remain largely unknown. In this study, we identified several BRD4 inhibitors, including molibresib, a novel pexophagy inducer, via chemical library screening. Treatment with molibresib promotes loss of peroxisomes selectively, but not mitochondria, ER, or Golgi apparatus in HeLa cells. Consistently, depletion of BRD4 expression also induced pexophagy in RPE cells. In addition, the inhibition of BRD4 by molibresib increased autophagic degradation of peroxisome ATG7-dependency. We further found that molibresib produced reactive oxygen species (ROS), which potentiates ATM activation. Inhibition of ROS or ATM suppressed the loss of peroxisomes in molibresib-treated cells. Taken together, our data suggest that inhibition of BRD4 promotes pexophagy by increasing ROS and ATM activation.

## 1. Introduction

Peroxisomes—in cooperation with mitochondria—play multiple roles in diverse metabolic events, including the β-oxidation of fatty acids, lipid biogenesis, and reactive oxygen species (ROS) metabolism [1]. In addition, peroxisomes serve as a signaling platform for immune responses; thus, peroxisome dysregulation has been implicated in various human disorders, such as metabolic diseases, cancer, and neurodegenerative diseases [2]. As a dynamic organelle, the content, shape, and size of peroxisomes vary among different cell types. To maintain optimal peroxisome number and function, the quality and quantity of peroxisomes are tightly regulated by various peroxisomal proteins, known as peroxisomal biogenesis factors (PEXs) [1]. Peroxisomal proteins are synthesized in the cytosol and localized to the peroxisome by the recognition of specific peroxisomal targeting signals (PTS1, 2 and mPTS) [3,4,5]. As a signal sequence, most PEXs have a conserved tripeptide sequence at their carboxyl terminus (PTS1). A few peroxisomal matrix proteins are recognized based on the presence of a PTS2 motif near the N-terminus. In addition, peroxisomal membrane proteins (PMPs) are targeted to the peroxisome by the membrane PTS (mPTS) [3,4,5]. Therefore, the presence of diverse *PEX* mutants results in peroxisomal biogenesis disorders (PBDs), including Zellweger syndrome, which is characterized by defective peroxisome assembly and function [6,7].

Autophagy is a catabolic process responsible for the degradation of cellular components. Under stressful conditions, autophagosomes engulf harmful cytosolic proteins or damaged organelles [8]. The cytosolic form of microtubule-associated protein 1A/1B-light chain-3-I (LC3-I) is converted to LC3-II, which is recruited to the autophagosomal membranes. Autophagosomes fuse with lysosomes to degrade the cellular components [9]. Various autophagy-related proteins (ATGs), including ATG5 and ATG7, mediate canonical autophagic flux [10]. Pexophagy, which refers to the selective autophagic degradation of peroxisomes, promotes the elimination of dysfunctional or superfluous peroxisomes under stress conditions such as starvation and hypoxia [11,12]. Recent studies on pexophagy have shown that the ubiquitination of membrane proteins in peroxisomes is required for pexophagy. Ataxia-telangiectasia mutated (ATM) kinase is activated in response to excessive ROS production and phosphorylates PEX5, which promotes its ubiquitination. Subsequently, ubiquitinated PEX5 is recognized by p62, a phenomenon that results in the recruitment of autophagosomes [12,13].

Although we and other groups have recently reported several pexophagy inducers, such as 1,10-phenanthroline, 2,2’-dipyridyl, and 3-amino-1,2,4-triazole (3-AT) [14,15,16], the molecular mechanisms underlying pexophagy in mammals are still poorly understood. In this study, we newly identified several inhibitors for bromodomain and extra-terminal domain (BET) family proteins (BRDs) as pexophagy inducers. Inhibition of BRD4 via genetic knockdown or a chemical inhibitor, molibresib, promoted the loss of peroxisomes, a phenomenon that was completely reversed by the inhibition of ROS and ATM.

## 2. Materials and Methods

### 2.1. Reagents

Libraries containing PPI inhibitor (L9400) and ubiquitination compound (L8600)— for drug screening—were obtained from TargetMol (Boston, MA, USA). Molibresib (10676), I-BET151 (11181) and dBET1 (18044) were obtained from Cayman Chemical Co. (Ann Arbor, MI, USA). *N*-acetyl-L-cysteine (NAC) (A9165), Bafilomycin A1 (B1793), and KU55933 (SML1109) were purchased from Sigma-Aldrich (St. Louis, MO, USA). Hoechst 33,342 (H3570) was obtained from Thermo Fisher Scientific (Waltham, MA, U.SA.) and DRAQ5 (ab108410) was obtained from Abcam (Cambridge, U.K.). Short interfering RNA (siRNA) targeting *BRD4* (#1, 5′-AGAUUGAAAUCGACUUUGAUU-3′ and #2, 5′-UGAGCACAAUCAAGUCUAAUU-3′), *NBR1* (5′-GAACGUAUACUUCCCAUUGUU-3′), *p62*/*SQSTM1* (5′-GCAUUGAAGUUGAUAUCGAUUU-3′), and scrambled siRNA (5′-CCUACGCCACCAAUUUCGU-3′) were synthesized by Genolution (Seoul, Korea).

### 2.2. Cell Culture and Establishment of Stable Cell Lines

hTERT RPE-1 cells stably expressing monomeric red fluorescent protein (mRFP)-enhanced green fluorescent protein (EGFP)-serine-lysine-leucine (SKL) (RPE/mRFP-EGFP-SKL) were kindly provided by Dr. Raekil Park (GIST, Korea) [15]. HeLa cells were obtained from the American Type Culture Collection (Manassas, VA, USA.). ATG7-knockout (KO) HeLa cells generated using the CRISPR/Cas9 system were kindly provided by Dr. T. Kanki (Niigata University, Niigata, Japan) [17]. All cells were cultured at 37 °C in an incubator containing a 5% CO_2_ atmosphere and maintained in DMEM supplemented with 10% FBS (SH30243.01 and SH30084.03; Hyclone, Logan, UT, USA.) and 1% penicillin-streptomycin (15140122; Invitrogen, Waltham, MA, USA.). To generate stable cell lines, HeLa cells were transfected with pmTurquoise2-Peroxi (HeLa/Peroxi), pmTurquiose2-Mito (HeLa/Mito), pmTurquoise2-Golgi (HeLa/Golgi), and pmTurquoise2-ER (HeLa/ER) and pHyPer-PTS1 (HeLa/pexo-HyPer), pHyPer-mitochondria (HeLa/mito-HyPer), pHyPer-cytosol (HeLa/cyto-HyPer) using Lipofectamine 2000 (11668019; Invitrogen), in accordance with the manufacturer’s protocol. Stable transfectants were established by growth in selection medium containing 1 mg/mL G418 (10131035; Invitrogen) for 7 days. After seeding the individual cells, stable clones were identified under a fluorescence microscope (IX71; Olympus, Tokyo, Japan).

### 2.3. Determination of Pexophagic Cells

To quantify cells undergoing pexophagy, HeLa/Peroxi and HeLa cells were grown on a glass cover slip and treated with molibresib at different time points. The cells were then washed with PBS (LB 001–02; WELGENE, Daegu, Korea), fixed with 4% paraformaldehyde (P2031; Biosesang, Seongnam, Korea) for 20 min, and stained with the indicated dyes or antibodies. Punctate peroxisome structures were visually determined using confocal microscopy. The number of peroxisomal structural (ABCD3) puncta was quantified using ImageJ (NIH, Bethesda, MD, USA), and the experiments were performed at least three times for each condition, as indicated in each Figure. Peroxisome number was calculated by dividing the number of peroxisomal structures by the cell volume. The same method was used to quantify the number of ABCD3 puncta.

### 2.4. Confocal Microscopy

Stable cell lines were grown on a glass cover slip and treated with molibresib for different times, washed with PBS, and fixed with 4% paraformaldehyde for 20 min. Fluorescent images of subcellular localization and morphology of peroxisomes were obtained using a confocal laser scanning microscope (LSM 800; Carl Zeiss, Oberkochen, Germany).

### 2.5. Western Blotting

Cell lysates were prepared in 2× Laemmli sample buffer (1610737; Bio-Rad). Total protein quantity was measured using the Bradford dye (5000001; Bio-Rad) in accordance with the manufacturer’s instructions. The samples were then separated by SDS polyacrylamide gel electrophoresis and transferred to a PVDF membrane (1620177; Bio-Rad). After blocking with 4% skim milk (BD Bioscience, 90002–594) prepared in TBST [Tris base (T9200; GenDEPOT, Baker, TX, USA), NaCl (G0610; GenDEPOT), and Tween^®®^ 20 (P7949; Sigma-Aldrich)], the membrane was incubated with the indicated primary antibodies. Anti-BRD4 (ab128874), anti-PEX10 (ab196827), and anti-phospho-ATM (ab81292) antibodies were purchased from Abcam; anti-LC3 (NB100-2220) antibody was purchased from NOVUS Biologicals (Centennial, CO, USA); anti-ATM (GTX70103) was purchased from GeneTex (Irvine, CA, USA); and anti-ACTA1 (MAB1501) was purchased from Sigma. For protein detection, membranes were incubated with HRP-conjugated secondary antibodies (7076S and 7074S; Cell Signaling Technology, Danvers, MA, USA).

### 2.6. ROS Measurement

Cellular ROS levels were determined using the fluorescent dye dichloro-dihydro-fluorescein diacetate (DCFH-DA), which was obtained from Thermo Fisher Scientific (C2938). Briefly, HeLa cells were seeded in 96-well plates and treated with chemicals in the presence or absence of NAC. After 48 h of treatment, cells were incubated with DCFH-DA for 10 min. Fluorescence measurements were performed using ImageJ, and the experiments were performed at least three times for each condition, as indicated in each Figure. Fluorescence intensity was calculated using 300 cells and based on the time of each counting.

### 2.7. ROS Measurement with HyPer Protein

ROS levels were assessed using a HyPer protein system. Approximately 10^4^ HeLa cells expressing peroxisomal HyPer, mitochondrial HyPer or cytosolic HyPer were seeded on 96-well plates. After 24 h, the cells were further treated with individual chemicals in the presence or absence of NAC. After 48 h of treatment, fluorescence measurements were performed using ImageJ, and the experiments were performed at least three times for each condition, as indicated in each Figure. Fluorescence intensity was calculated using 300 cells and based on the time of each counting.

### 2.8. Statistical Analysis

Data were obtained from at least three independent experiments and are presented as means ± S.E.M. Statistical evaluation of the results was performed using one-way ANOVA. * *p* < 0.05, ** *p* < 0.01, and *** *p* < 0.001 were considered significant, but *p* > 0.05 was considered as statistically non-significant.

## 3. Results

### 3.1. Inhibition of BRD4 Promotes Pexophagy

Pexophagy is important for peroxisome quality control. To identify novel pexophagy regulators, we employed a cell-based high-content screening system in hTERT RPE-1 cells stably expressing mRFP-EGFP-SKL protein (RPE/mRFP-EGFP-SKL) [15]. Using this cell line, we screened a chemical library consisting of ubiquitination-related small chemicals and protein–protein interaction inhibitors, and identified several pexophagy-inducing candidates, including molibresib, I-BET151, and dBET1, which commonly target the BET family proteins (BRDs), i.e., BRD2, BRD3, BRD4, and BRDT. BRDs are major transcriptional regulators [18]. To verify the screening results, RPE/mRFP-EGFP-SKL cells were treated with molibresib, I-BET151, and dBET1, and pexophagy was monitored. As shown in Figure 1A, yellow puncta that represent peroxisomes were visible owing to the merging of [EGFP(+)/mRFP(+)]-double positive fluorescence in the control cells, whereas the number of red only puncta with a reduced green signal [EGFP(-)/mRFP(+)] was highly increased in the BRD inhibitor-treated cells (Figure 1A). To further examine organelle-selective autophagy, we monitored other organelles, such as the ER, Golgi apparatus, and mitochondria, in molibresib-treated cells. HeLa cells stably expressing pmTurquiose2-Mito, pmTurquiose2-ER, pmTurquiose2-Golgi, or pmTurquiose2-Peroxi were treated with molibresib for 72 h, and the cellular organelles were monitored. Notably, among the observed organelles, the number of peroxisomes were highly reduced, whereas that of mitochondria, ER, and Golgi apparatus was not decreased after treatment with molibresib (Figure 1B). As molibresib, I-BET151, and dBET1 commonly target BRD4, we further addressed the effect of BRD4 inhibition on pexophagy by depleting BRD4 via RNA interference. Consistent with the results of chemical inhibition, the number of red puncta of mRFP-EGFP-SKL was significantly increased by depletion of BRD4 (Figure 1C,D), indicating that inhibition of BRD4 induces pexophagy in RPE cells. Taken together, these results suggest that molibresib-mediated BRD4 inhibition selectively induces loss of peroxisomes.

### 3.2. Loss of ATG7 Blocks Molibresib-Induced Peroxisomal Autophagy in HeLa Cells

We observed the loss of peroxisomes in molibresib-treated cells. As pexophagy refers to the selective degradation of peroxisomes via autophagy, we examined the effect of molibresib on autophagy in HeLa cells. The conversion of LC3-I to LC3-II is widely used to examine autophagy activation. As shown in Figure 2A,B, the levels of LC3-II were highly increased in molibresib-treated cells (Figure 2A), and LC3-II were higher in cells cotreated with molibresib and a lysosome fusion inhibitor (bafilomycin A1) than that of bafilomycin-single treated control cells. In addition, loss of peroxisome was completely blocked by bafilomycin in molibresib-treated cells (Figure 2B). These data indicate that molibresib increases autophagic flux in HeLa cells. We next confirmed the effect of autophagy inhibition in ATG7-knockout (KO) cells. Wild type and ATG7-KO HeLa cells were treated with molibresib and loss of peroxisomes was addressed. Consistently, molibresib highly induces loss of peroxisomes in wild type HeLa cells (Figure 2C,D); however, it was completely blocked in ATG7-KO cells (Figure 2C). Moreover, we also examined the levels of peroxisomal membrane protein PEX10 and observed that knockout of ATG7 efficiently blocked the decrease of PEX10 in molibresib-treated cells (Figure 2D). Taken together, our results suggest that molibresib promotes pexophagy by an ATG7-dependent canonical autophagic pathway. Both NBR1 and p62 are involved in pexophagy as an autophagy receptor for peroxisome [1,12]. We further investigated the effect of both autophagy receptors in molibresib-treated cells. As shown in Figure 2E, deletion of either NBR1 or p62 significantly suppressed loss of peroxisome in molibresib-treated cells, suggesting that NBR1 and p62 act as autophagy receptors in pexophagy in molibresib-treated cells (Figure 2E).

### 3.3. Molibresib Promotes Pexophagy by Increasing ROS and ATM Activation

As BET proteins have multiple functions, BET inhibitors also regulate various events, including cellular stress. Recently, we and other groups have shown that excessive ROS promotes pexophagy [12,16,19]. Thus, we investigated the role of oxidative stress on pexophagy in molibresib-treated cells. As shown in Figure 3A, treatment of molibresib significantly increased the cellular ROS levels, a phenomenon that was completely blocked by NAC, a ROS scavenger (Figure 3A). To examine peroxisome-specific ROS, we employed a peroxisomal redox-sensitive fluorescent protein, HyPer-PTS1 [19]. HeLa/pexo-HyPer cells were treated with molibresib with or without NAC, and the fluorescence intensity of peroxisomal HyPer protein was measured. Consistently, molibresib significantly enhanced the fluorescence, which was also blocked by NAC, indicating that the molibresib increased peroxisomal H_2_O_2_ levels in HeLa cells (Figure 3B). Interestingly, we found that inhibition of ROS by NAC suppressed the loss of peroxisome by molibresib in HeLa cells (Figure 3C). Ubiquitination of the membrane proteins of target organelles is required for selective autophagy [13]. According to this hypothesis, pexophagy by ROS is induced by ubiquitination of PMPs, such as ABCD3 [11]. Zhang et al. showed that ATM phosphorylates PEX5, which recruits autophagosomes during oxidative stress [1,12]. Therefore, we examined the role of ATM in molibresib-mediated pexophagy. As shown in Figure 4A, molibresib slightly upregulated ATM expression, but markedly increased its phosphorylation in HeLa cells (Figure 4A). Additionally, this enhanced phosphorylation of ATM by molibresib was completely abolished by NAC (Figure 4A), indicating that the increased ROS levels in response to molibresib treatment mediates the activation of ATM. Therefore, we further evaluated ATM activation on molibresib-induced pexophagy. HeLa cells were treated with molibresib with or without KU55933, an ATM inhibitor, and pexophagy was examined. The results showed that the loss of peroxisomes in response to molibresib exposure was restored upon ATM inhibition (Figure 4B). Taken together, these results suggest that molibresib induces the loss of peroxisomes by increasing the ROS levels and ATM activation.

## 4. Discussion

Peroxisomes are highly oxidative organelles because of their function in generating and scavenging hydrogen peroxide. They are involved in the catabolism of various fatty acids and in the biosynthesis of several lipids. Although the quality control of peroxisomes by pexophagy is important for understanding the pathology of PBDs, the precise molecular mechanism underlying pexophagy is largely unknown. In this study, we identified molibresib as a strong inducer of pexophagy. Molibresib is a small molecule inhibitor of BET family members [20]. BRD4, a member of BET family protein, recruits transcriptional regulatory components to acetylated histones in order to positively regulate various processes, such as cell growth and cell cycle via modulation of gene expression [18]. Notably, recently, Sakamaki et al. showed that BRD4 functions as a transcriptional repressor of autophagy and lysosomal activity [21]. They showed that depletion of BRD4, but not that of BRD2 or BRD3, results in autophagy activation by upregulating multiple autophagy regulatory genes that encode proteins involved in autophagosome formation, autophagy cargo recruitment, and autophagosome–lysosome fusion in vitro and in vivo [21]. In addition, the enhanced autophagy by BRD4 knockdown confers cell survival under starvation conditions by providing a nutrient source [21]. Furthermore, Wakita et al. also suggested that a BET family protein degrader promotes senolysis by upregulating gene expression of autophagy in senescent cells [22]. BRD4 represses expression of autophagy and lysosomal genes by interacting with a euchromatic histone-lysine N-methyltransferase 2 (EHMT2) [21,23]. Interestingly, we and other groups have previously reported that BIX-01294, an inhibitor of EHMT2, strongly induces autophagy in different cells by upregulating autophagy-regulatory genes, including *BECN1*, *LC3*, and *WIPI*, and increasing the ROS levels [24,25,26]. However, the role of BRD4 in pexophagy has not been elucidated. To identify novel pexophagy regulators, we initially screened a chemical library in RPE/mRFP-EGFP-SKL cells. The screening results showed that the inhibition of BRD4 by treatment with different chemical inhibitors, including molibresib, I-BET151, dBET1, or by RNAi knockdown, resulted in an increase in the number of red puncta, representing pexophagy in RPE cells (Figure 1A,C). The loss of peroxisomes in response of molibresib treatment could be attributed to either increased degradation of peroxisomes or decreased peroxisome biogenesis. Hence, we examined the effect of molibresib on loss of peroxisomes in ATG7-knockout cells and observed that inhibition of autophagy by depletion of ATG7 almost completely blocked molibresib-induced peroxisome loss in HeLa cells (Figure 2). Nonetheless, further studies regarding the role of molibresib in peroxisome biogenesis are needed, and our findings suggest that molibresib promotes the autophagy-dependent clearance of peroxisomes.

Interestingly, we also found that the number of peroxisomes, but not that of mitochondria, ER, and Golgi, was remarkably reduced upon treatment with molibresib in HeLa cells (Figure 1B). It has been previously reported that BRD4 depletion sustains mTOR activity, which can induce non-selective autophagy during starvation [21]. However, our results further suggest that molibresib-mediated inhibition of BRD4 strongly promoted pexophagy rather than non-selective autophagy in HeLa cells. Consistent with these observations, we additionally confirmed that molibresib increased the peroxisomal ROS as well as mitochondrial and cytosolic ROS by monitoring fluorescence intensity of HyPer protein (Figure 3 and Appendix A). Notably, we showed that in response to excessive ROS levels, ATM induces the phosphorylation of PEX5 to recruit autophagy adaptors for enabling pexophagy [12,19]. In addition, we found that molibresib upregulated ATM expression and its phosphorylation (Figure 4A). Although our results highlight the role of ROS-ATM in molibresib-induced pexophagy, it was reported that inhibition of BET epigenetically upregulates expression of genes associated with cellular redox status and cell death, such as *thioredoxin-interacting protein (TXNIP)* [27]. TXNIP participates in metabolic reprogramming and induces oxidative stress in various cancer cells [28]. Thus, the mechanism by which molibresib can generate excessive ROS and modulation of expression of oxidative stress-associated genes, including *TXNIP* and *ATM*, by BRD4 requires further investigation. Various metabolic profiles observed in cancer cells have suggested that cancer is a metabolic disease [29]. According to this hypothesis, recent studies have suggested that several peroxisome metabolic functions, such as lipid β-oxidation and ether phospholipid biosynthesis, are dysregulated in cancers [30,31]. In addition, the expression of various peroxisomal-related genes, such as *DEPP* and *PEX16*, is highly upregulated in several cancer types, including melanoma, colon, lung, ovarian, and prostate cancers [30,31,32], suggesting that peroxisomes are involved in tumorigenesis. Walter et al. reported that HIF-2α can trigger pexophagy under hypoxia stress. Activation of HIF-2α, which is overexpressed in clear cell renal cell carcinoma (ccRCC) increases peroxisome turnover via pexophagy [33]. HIF-2α-mediated loss of peroxisomes alters some lipid metabolism, a phenomenon that may promote malignant activity in ccRCC [33,34]. As BRD4 is frequently upregulated in various cancers and plays multiple oncogenic roles, including promotion of cell proliferation and metastasis, selective inhibitors of BRD4 are key molecules that are investigated using preclinical and clinical models of solid tumors [35,36]. In this study, we demonstrated that molibresib-mediated inhibition of BRD4 strongly promoted pexophagy in cancer cells. Therefore, further studies on the role of pexophagy in the regulation of peroxisome homeostasis will be helpful for understanding the role of peroxisomes in tumorigenesis, and, by extension, anti-cancer drug development.

## Figures and Tables

**Figure 1 cells-11-02839-f001:**
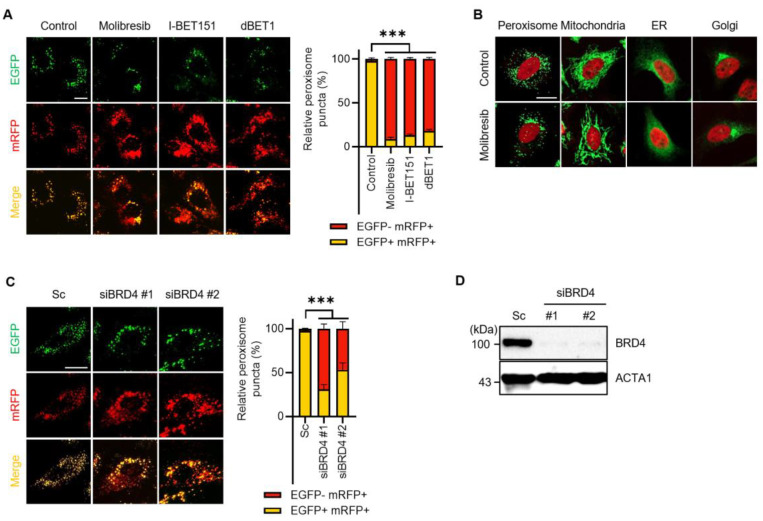
**Inhibition of BRD4 promotes pexophagy in RPE cells.** (**A**) RPE/mRFP-EGFP-SKL cells were treated with inhibitors of BRD4 [molibresib (10 µM), I-BET151 (10 µM), dBET1 (10 µM)] for 72 h. The cells were fixed for imaging under a fluorescence microscope. The numbers of cell with EGFP(+)- and mRFP(+)-labeled autophagosomes or EGFP(-) and mRFP(+)-labeled autolysosomes, which displayed the peroxisomal reporter mRFP-EGFP-SKL due to lysosomal delivery, were counted and are presented as percentages; (**B**) HeLa cells stably expressing pmTurquiose2-Peroxi, pmTurquiose2-Mito, pmTurquiose2-ER, or pmTurquiose2-Golgi were treated with molibresib (10 µM) for 72 h and stained with DRAQ (red). Cellular organelles were imaged by confocal microscopy. The scale bar indicates 20 µm; (**C**,**D**) RPE/mRFP-EGFP-SKL cells were transiently transfected with siRNA targeting *BRD4* (siBRD4), and then, EGFP and mRFP fluorescence was imaged and quantified (**C**); reduced expression of BRD4 by siRNA was verified by Western blotting (**D**). (Data indicate means ± S.E.M. *** *p* < 0.001).

**Figure 2 cells-11-02839-f002:**
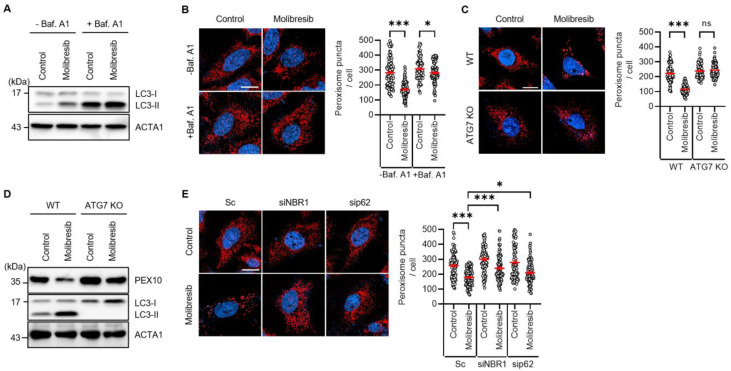
**Loss of ATG7 blocks molibresib-induced pexophagy in HeLa cells.** (**A**,**B**) HeLa cells pre-treated with molibresib (10 µM) for 48 h were further incubated with or without bafilomycin A1 (Baf. A1, 25 nM) for 6 h. Then, the cells were harvested to analyze by Western blotting with indicated antibodies (**A**); the cells were stained with anti-ABCD antibody (red) and Hoechst 33,342 (blue). The number of peroxisomes per cell was calculated by assessing > 100 cells (**B**); (**C**) wild-type and ATG7-knockout HeLa cells were treated with molibresib (10 µM) for 48 h. The cells were stained with anti-ABCD3 antibody (red) and Hoechst 33,342 dye (blue). The number of peroxisomes per cell was calculated by assessing approximately 100 cells; (**D**) wild-type and ATG7-knockout HeLa cells were treated with molibresib (10 µM) for 48 h, and then harvested for analysis by Western blotting with the indicated antibodies; (**E**) HeLa cells were transiently transfected with scrambled siRNA (Sc) or validated siRNA targeting for NBR1 (siNBR1) or p62 (sip62). After 24 h, the cells were further treated with molibresib (10 µM) for 48 h and stained with anti-ABCD3 antibody (red) and Hoechst 33,342 dye (blue). The number of peroxisomes per cell was calculated by assessing approximately 100 cells; the scale bar indicates 20 µm. (Data indicate means ± S.E.M. * *p* < 0.05, *** *p* < 0.001, ns: not significant).

**Figure 3 cells-11-02839-f003:**
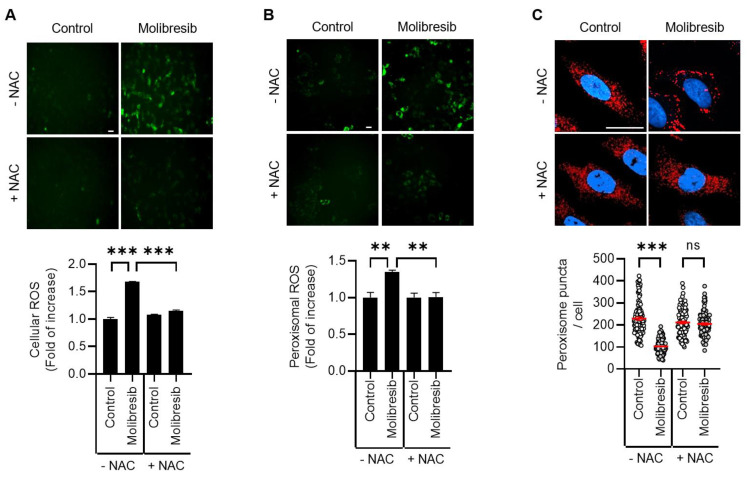
**Inhibition of BRD4 promotes pexophagy by increasing the ROS levels in HeLa cells.** (**A**) HeLa cells were treated with molibresib (10 µM) with or without NAC (1 mM) for 48 h. Then, the cells were incubated with fluorescent DCFH-DA dye (10 µM, 10 min). The cells were imaged (scale bar 50 µm), and DCFH-DA fluorescence intensity was measured using image processing software ImageJ; (**B**) HeLa/pexo-HyPer cells were treated with molibresib (10 µM) with or without NAC (1 mM) for 48 h. The level of peroxisomal H_2_O_2_ was imaged (scale bar, 50 μm) and measured using the fluorescence intensity of HyPer-PTS1; (**C**) HeLa cells were treated with molibresib (10 µM) with or without NAC (1 mM) for 48 h. Then, the cells were immunostained with anti-ABCD3 antibody (red) and Hoechst 33,342 dye (blue) to count the number of peroxisomes in cells. Representative cell images are presented (C, scale bar 20 µm). The experiments were repeated at least three times. (Data indicate means ± S.E.M. ** *p* < 0.01, *** *p* < 0.001, ns: not significant).

**Figure 4 cells-11-02839-f004:**
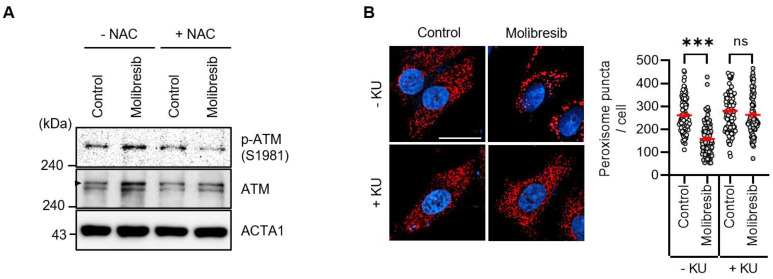
**Inhibition of BRD4 activates ATM to promote pexophagy in HeLa cells.** (**A**) HeLa cells were treated with molibresib (10 µM) with or without NAC (1 mM) for 48 h. The cells were harvested and analyzed by Western blotting with the indicated antibodies; (**B**) HeLa cells were treated with molibresib (10 µM) with or without KU55933 (10 µM) for 48 h. Then, the cells were immunostained with anti-ABCD3 antibody (red) and Hoechst 33,342 dye (blue) to count the peroxisomes. Representative cell images are presented (scale bar 20 µm). The experiments were repeated at least three times. (Data indicate means ± S.E.M. *** *p* < 0.001, ns: not significant).

## Data Availability

Not applicable.

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
