# Peer review of "Inhibition of BRD4 Promotes Pexophagy by Increasing ROS and ATM Activation"

_cells, 2022, doi:10.3390/cells11182839_

Round 1

Reviewer 1 Report

YH Kim and DS Jo et al., characterized Inhibition of BRD4 esp. molibresib as a novel pexophagy inducer. They also demonstrated molibresib treatment in only endorses peroxisomal loss but not other organelles such as mitochondria, ER or golgi bodies.   Although, this is an interesting observation to the research field of ROS and selective autophagy (pexophagy), there are several points the authors need to address to improve their study.

1.        As authors stated BRD4 inhibitors esp. molibresib induces ROS. However, I did not find in result or discussion section that how did molibresib induces ROS.

2.        Not only peroxisome but mitochondria as well as ER is also the source of ROS (M. Schrader et al.,BBA 2006; RK Dutta et al., Biofactors 2021). It will be more appreciated, if the authors check source of ROS by molibresib treatment.

3.        Result section 3.1 and figure 1 description is not matching.

4.        In Fig. 2A and line no. 189-191, authors stated that “The conversion of LC3-I to LC3-II is widely used to examine autophagy activation. As shown in Figure 2A, the levels of LC3-II were highly increased in molibresib-treated cells.” This statement did not clarify the activation of autophagy. Although it is usually considered that the conversion of soluble LC3-I to lipid bound LC3-II is associated with the formation of autophagosomes, The activation of LC3-I to LC3-II may then be necessary but not sufficient to trigger cell autophagy (P Giménez-Xavier et al.,ijmm 2008).Hence, I request author to check autophagy receptor protein p62. Morever, bafilomycin treatment also induce LC3-II, did the authors clarify how a lysosome fusion inhibitor, bafilomycin induce autophagy.

5.        What is the pexophagy receptor? Is it NBR1, p62 or other protein?

6.        Authors mentioned many times that they analyze BRD4 inhibitors, including molibresib a novel pexophagy inducer via chemical library screening. I did not find any figure or any cited paper where they do chemical library screening.

7.        Page 1, line no. 35-37, please cite the statement Peroxisomal proteins are synthesized in the cytosol and localized to the peroxisome by the recognition of specific peroxisomal targeting signals (PTS1, 2 and mPTS).

8.        Fig 1 and Fig 4 is same

Author Response

Response to Reviewers' comments: (Reviewer #1)

YH Kim and DS Jo et al., characterized Inhibition of BRD4 esp. molibresib as a novel pexophagy inducer. They also demonstrated molibresib treatment in only endorses peroxisomal loss but not other organelles such as mitochondria, ER or golgi bodies.   Although, this is an interesting observation to the research field of ROS and selective autophagy (pexophagy), there are several points the authors need to address to improve their study.

Q1. As authors stated BRD4 inhibitors esp. molibresib induces ROS. However, I did not find in result or discussion section that how did molibresib induces ROS.

Response 1: We appreciate this opportunity to improve our manuscript. Until now, how molibresib can induce excessive ROS is not clear yet. Although it has been reported that BRD4 inhibitor, JQ1 decreased ROS production in H2O2-treated prostate cancer cells (Hussong M et al, Cell Death Dis, 2014), recent evidences have suggested that inhibition of BET up-regulates thioredoxin-interacting protein (TXNIP), which promotes oxidative stress and anti-proliferation in breast cancer cells (Alluri et al., Cell Research, 2014; Zhou Y et al., BMC Cancer 2018). Since molibresib is a potent inhibitor for BET proteins including BRD4, it may possible that molibresib regulates expression of genes associated with oxidative stress. We have described the hypothesis in discussion part and will further evaluate this possibility.

Q2. Not only peroxisome but mitochondria as well as ER is also the source of ROS (M. Schrader et al., BBA 2006; RK Dutta et al., Biofactors 2021). It will be more appreciated, if the authors check source of ROS by molibresib treatment.

Response 2: Thank you for your valuable comments. There are several sources for ROS generation in cells. To further evaluate ROS source, we have established HyPer systems with mitochondria-HyPer, peroxisome-HyPer and cytosolic-HyPer. As shown in reviewer’s Figure1, Not only fluorescence of pexo-HyPer, but also fluorescence of mito-HyPer and cyto-HyPer were increased by treatment of molibresib. However, the fluorescence of pexo-HyPer was slightly higher than that of mito-HyPer. Since ROS can cross cellular organelles, our system could not distinguish different ROS sources.

Reviewer Figure 1: HeLa cells stably expressing mitochondrial HyPer protein (mito-HyPer), peroxisomal HyPer protein (pexo-HyPer), or cytosolic HyPer (cyto-HyPer) were treated with molibresib (10 µM) for 48 hours. Then the fluorescence of HyPer proteins was imaged and measured by imageJ program.

Q3. Result section 3.1 and figure 1 description is not matching.

Response 3: We sincerely apologize for our huge mistake. The Figure1 was replaced by Figure4 during the submission stage. We have added the Figure1 correctly in Revised Figure1.

Q4-1. In Fig. 2A and line no. 189-191, authors stated that “The conversion of LC3-I to LC3-II is widely used to examine autophagy activation. I request author to check autophagy receptor protein p62. Q4-2. Moreover, bafilomycin treatment also induce LC3-II, did the authors clarify how a lysosome fusion inhibitor, bafilomycin induce autophagy.

Response 4-1: We appreciate your suggestion. According to this generous suggestion, we have examined the effect of p62 deletion in molibresib-induced pexophagy. Knock down of p62 by its siRNA showed significantly suppressed loss of peroxisome in molibresib-treated cells. We added the result in revised Figure 2E.

Response 4-2: Bafilomycin A1 is not autophagy inducer but inhibitor. Bafilomycin disrupts autophagy flux by inhibiting fusion between autophagosomes and lysosomes. Thus, treatment with bafilomycin A1 induces accumulation of LC3-II with autophagosomes (Klionsky DJ et al. Autopahgy 2008).  

Q5. What is the pexophagy receptor? Is it NBR1, p62 or other protein?

Response 5: We appreciate your suggestion. As NBR1 and p62 have been identified as a major pexophagy receptor, we further addressed the autophagy receptors in molibresib-induced pexophagy. Notably, depletion of either NBR1 or p62 significantly suppressed the molibresib-induced loss of peroxisome, suggesting that both NBR1 and p62 are involved in the molibresib-induced pexophagy. We added the results in revised Figure 2E.

Q6. Authors mentioned many times that they analyze BRD4 inhibitors, including molibresib a novel pexophagy inducer via chemical library screening. I did not find any figure or any cited paper where they do chemical library screening.

Response 6: We sincerely apologize for this confusion. The screening result was described in Figure1, which was missed. We correctly added the Figure1 in revised manuscript. 

Q7. Page 1, line no. 35-37, please cite the statement Peroxisomal proteins are synthesized in the cytosol and localized to the peroxisome by the recognition of specific peroxisomal targeting signals (PTS1, 2 and mPTS).

Response 7: We appreciate your comment. According to this generous suggestion, we added a reference (#3 and #4) for the sentence.

Q8. Fig 1 and Fig 4 is same

Response 8: We sincerely apologize again for our huge mistake. It is a related question with #3. The Figure1 was replaced by Figure4 during the submission stage. We have added the Figure1 correctly in revised Figure1.

Reviewer 2 Report

The manuscript entitled "Inhibition of BRD4 promotes pexophagy by increasing ROS and ATM activation." has been submitted as an original article by Kim et al.

The manuscript presents interesting data concerning the role of the BET-family protein BRD4 in pexophagy regulation in mammalian cells.

In general, the experiments are sound and the conclusions are supported by the data.

Minor point:

The authors should add molecular weight marker to the Western Blot figures.

Author Response

Response to Reviewers' comments: (Reviewer #2)

The manuscript entitled "Inhibition of BRD4 promotes pexophagy by increasing ROS and ATM activation." has been submitted as an original article by Kim et al. The manuscript presents interesting data concerning the role of the BET-family protein BRD4 in pexophagy regulation in mammalian cells. In general, the experiments are sound, and the conclusions are supported by the data.

Q1. Minor point: The authors should add molecular weight marker to the Western Blot figures.

Response 1: We appreciate this opportunity to improve our manuscript. According to the reviewer’s suggestion, we have added the makers for molecular weight in revised Figures.

Reviewer 3 Report

The authors present the manuscript entitled "Inhibition of BRD4 Promotes Pexophagy by Increasing ROS 2 and ATM Activation". The manuscript is well-written and adequately presented.

However several issues should be resolved prior acceptance. 

1. The materials and methods section should be presented in more details. For instance, for the peroxisomal ROS measurement, please specify the duration of the treatment.

2. P values bellow 0.05 should also be considered significant. Please explain in the Statistical analysis section (under Materials and methods) what non significant means (p above 0.05?)

3. Please specify the concentration and time-duration of BafA1 treatment (all relevant figure legends).

4. The authors only used ATG7 KO cells to show the autophagy mechanism. The authors should investigate the peroxisome puncta also upon BafA1 treatment (Figure 2 C should be repeated with BafA1).

Author Response

Response to Reviewers' comments: (Reviewer #3)

The authors present the manuscript entitled "Inhibition of BRD4 Promotes Pexophagy by Increasing ROS and ATM Activation". The manuscript is well-written and adequately presented. However, several issues should be resolved prior acceptance. 

Q1. The materials and methods section should be presented in more details. For instance, for the peroxisomal ROS measurement, please specify the duration of the treatment.

Response 1: We appreciate this opportunity to improve our manuscript. According to the reviewer’s suggestion, we described more detail for the experiments.

Q2. P values bellow 0.05 should also be considered significant. Please explain in the Statistical analysis section (under Materials and methods) what non-significant means (p above 0.05?)

Response 2: According to the reviewer’s generous suggestion, we described the p value in Material and Method. Yes, non-significant means above 0.05 in our experiments.

Q3. Please specify the concentration and time-duration of BafA1 treatment (all relevant figure legends).

Response 3: We have more described experimental conditions in Figure legends.

Q4. The authors only used ATG7 KO cells to show the autophagy mechanism. The authors should investigate the peroxisome puncta also upon BafA1 treatment (Figure 2 C should be repeated with BafA1).

Response 4: According to the reviewer’s generous suggestion, we further examined the effect of Baf.A1 and have added in the revised Figure 2B.

Round 2

Reviewer 1 Report

Thanks Authors for your pointwise response. Although you tried to answer the issues which I raised, I still not convinced with some of your response.  

1.      In response to Q. 2, you mentioned that not only peroxisome but also fluorescence intensity of mitochondria and cytosol were increased by treatment of molibresib. However you concluded that due to higher fluorescence intensity of pexo-HyPer, molibresib treatment induce ROS through peroxisome.

I request author to check ACOX1 level, as it is the first and a rate-limiting enzyme and major generator of ROS in peroxisome (Chen XF et al. EMBO Rep. 2018, Dutta RK et al Biofactors. 2021), to confirm the ROS generated by treatment of molibresib is through peroxisome.

2.      In response to Q. 4-1, I request authors to check the protein level of p62 or NBR1 to clarify the autophagy or specific autophagy activation by treatment of molibresib. Notably in response to Q.4-2, I want to know what the purpose of using bafilomycin. As treatment of bafilomycin as well as molibresib both induce LC3-II expression, then how can author conclude that one activate autophagy where as other inhibit autophagy.

3.       In response to Q.7, I want to know how peroxisomal protein are synthesized in cytosol ,  describe it or refer clear reference

Author Response

Please check an attached PDF file

Reviewer 3 Report

The authors fulfilled the requierments of this reviewer.

Author Response

The authors fulfilled the requierments of this reviewer.

Reply: Thanks for your helpful comments